# The Effect of Concentrate/Iron Ore Ratio Change on Agglomerate Phase Composition

**Mária Fröhlichová [1],\*, Dušan Ivanišin [2], Róbert Findorák [1], Martina Džupková [1] and Jaroslav Legemza [1]**

[1]   Faculty of Materials, Metallurgy and Recycling, Technical University of Košice, 04200 Košice, Slovakia; robert.findorak@tuke.sk (R.F.); martina.dzupkova@tuke.sk (M.D.); jaroslav.legemza@tuke.sk (J.L.)
[2]   U.S. Steel Košice s.r.o, Vstupný areál, 04454 Košice, Slovakia; DIvanisin@sk.uss.com
\*   Correspondence: maria.frohlichova@tuke.sk; Tel.: +421-55-602-3152

**Abstract:** The work is focused on studying the influence of the ratio of concentrate to iron ore on the phase composition of the iron ore agglomerate. The concentrate has significantly higher iron content than used iron ore, and is a determining factor, which influences the richness of the batch and consequently, the richness of the agglomerate. The increased iron content in the agglomerate can be achieved by adjusting the raw material ratio in which iron ore materials are added to the agglomeration mixture. If the ratio is in favor of iron ore this reflects in lower iron content in the resulting agglomeration mixture. If the ratio is in favor of a concentrate, which is finer, the fraction share of less than 0.5 mm will be increased, the permeability of the batch will be reduced, the performance of the sintering belt will decrease and the presence of solid pollutants will increase. The possibility of concentrate replacement by iron-rich iron ore with granulometry similar to that of concentrate was experimentally verified. The effect of the concentrate replacement by the finer iron-rich ore was tested in a laboratory sintering pan. There were performed six sinterings, with gradually changing ratio concentrate/iron ore (C/O). The change in the ratio of concentrate to iron ore, does not cause the occurrence of new phases, only the change in their prevalence, which does not bring a significant change of the qualitative indicators of the compared agglomerates. Concentrate replacement by iron ore up to 50% was optimal from technological, quality, and environmental aspects.

**Keywords:** concentrate; iron ore; agglomerate; structure; phase analysis

## 1. Introduction

Improving the quality of the iron ore agglomerate and reducing the energy intensity of blast-furnace metallurgical processes are an important prerequisite for improving the technical and economic indicators of blast furnace work. The requirements for the quality of agglomerate, the basic metal-bearing part of the blast furnace batch, are constantly increasing. With the current raw material base, when the amount of fine-grained iron ore materials is increasing, this can only be achieved by intensifying the agglomeration process. A good quality agglomerate is characterized by a suitable iron content, high reducibility, good strength and low fine grain shares content prior to charging into blast furnace and high strength after reduction in the blast furnace shaft [1].

In general, a wide range of materials is required to produce the agglomerate. For this reason, their preparation requires a great deal of attention to ensure the correct quality of the agglomerate and its suitability for the blast furnace process.

The mixture required to produce the agglomerate is composed of different types of ores, concentrates and secondary raw materials. It is characterized by fluctuations in the physico-chemical

properties and chemical composition and therefore it requires a thorough homogenization. Homogenization of batch materials is a demanding process due to the use of large volumes and variety of raw materials. Therefore, great emphasis is placed on the knowledge of the qualitative impacts of the batch, as well as on managing the whole technology of agglomeration batch preparation and processing.

For all components of raw materials suitable for agglomerate production, one of the most important aspects of assessment is granulometry. The maximum grain size for the agglomeration process must not exceed 10 mm [2,3]. The most suitable grain size for the sintering process is below 5 mm, or up to 3 mm for some raw materials [4].

The demand for increased iron content in agglomerates has opened up a possibility for the applying higher amounts of concentrates, which are generally richer in total iron than normal iron ores. Their increased proportion in agglomeration batch, on the other hand, brings problems associated with the pre-treatment of the mixture. Such increased requirements on the packing of a mixture with a higher share of concentrates, regarding the pre-pelletizing quality and packing time, present some challenges for the optimization of the sintering process [5].

The batch composition affects not only the quality of the agglomerate but also the ecology of the production. In addition to agglomerate quality monitoring, attention must be paid to the environmental aspect of production. It is important to look for links between the various production parameters. The emissions load on the environment can be reduced by ensuring the appropriate quality of batch raw materials as well as regulating the selected parameters of sintering, especially the height of the sintered layer, quantity and speed of the sucked air, ratio of the concentrate to iron ore [6–8].

Complex physico-chemical processes take place during agglomeration production and no generally-valid rules have been derived which might predict the composition of the finished product. However, there are certain known relationships and dependencies, according to which the desired agglomerate can be produced. The final properties of the agglomerate are determined not only by its chemical composition but also by its mineralogical composition, i.e., phases [9]. Mineralogical phases may have different structural forms depending on the conditions of agglomeration formation. Basically, there are crystalline and glass phases. A relatively large proportion of the mass is in the form of solid solutions of varying composition, making it difficult to quantitatively and qualitatively assess the contribution of the individual phases to the quality of the agglomerate. In view of this, only general conclusions can be made. It follows that when making agglomerates, we must take into account not only the chemical composition of the input components of the batch, but also the physico-mechanical properties and technological parameters. In general, all the technological parameters affect the phase composition and structure of the agglomerate and, in turn, its mechanical properties as well.

The theoretical prediction of the parameters of sintering and the properties of the finished product, as well as the environmental impact, are influenced by a number of factors which influence each other. This study focuses on the one of these factors, i.e., the effect of agglomerate properties of the replacement of concentrate with iron ore which approximating concentrate with its composition and granulometry. For this purpose, experiments were designed and performed on a laboratory sintering pan.

## 2. Materials and Methods

A total of six iron ore raw materials were used in the experiments: commonly-used iron ores AR-1, AR-2, AR-3, AR-4, concentrate KC1, and iron ore AR-5 with fine granulometry. AR-5 is just produced raw material (as delivered). All iron ores come from Ukrainian mines. The ores used in the experiment were not specially selected; they make part of the batch for the production of agglomerate used in the operating conditions. The number of iron ores used in the experiments is comparable to those utilized in commercial practice.

The water content of these ores, as delivered, is up to 5%, which is the most efficient for transport. The chemical composition of iron ores used for the experiment is shown in Table 1.

**Table 1.** The Chemical composition of iron ores used for experiment.

| Iron Ores | Humidity | Fe | FeO | Fe$_2$O$_3$ | SiO$_2$ | CaO | MgO | Al$_2$O$_3$ | Mn | P |
|---|---|---|---|---|---|---|---|---|---|---|
| | (%) | | | | | | | | | |
| AR-1 | 3.63 | 57.58 | 0.48 | 82.66 | 15.13 | 0.08 | 0.21 | 0.97 | 0.020 | 0.035 |
| AR-2 | 4.16 | 60.89 | 0.37 | 86.57 | 8.86 | 0.22 | 0.37 | 1.41 | 0.026 | 0.093 |
| AR-3 | 4.20 | 60.39 | 0.52 | 85.38 | 11.07 | 0.07 | 0.21 | 0.90 | 0.027 | 0.028 |
| AR-4 | 4.82 | 61.12 | 1.09 | 86.59 | 8.69 | 0.77 | 0.37 | 1.07 | 0.054 | 0.034 |
| Iron Ores | Basicity | S | Na$_2$O | K$_2$O | TiO$_2$ | Pb | Zn | As | C | Cl |
| | (%) | | | | | | | | | |
| AR-1 | 0.018 | 0.016 | 0.117 | 0.047 | 0.037 | 0.002 | 0.005 | 0.001 | 0.047 | 0.20 |
| AR-2 | 0.058 | 0.016 | 0.243 | 0.120 | 0.057 | 0.001 | 0.003 | 0.001 | 0.217 | 0.07 |
| AR-3 | 0.024 | 0.015 | 0.154 | 0.053 | 0.038 | 0.001 | 0.001 | 0.001 | 0.070 | 0.21 |
| AR-4 | 0.117 | 0.021 | 0.055 | 0.054 | 0.034 | 0.001 | 0.001 | 0.001 | 0.078 | 0.32 |

The most important aspect of the chemical composition of these iron ores is their iron content. In the experiment were used ores, which iron content is standard for Ukrainian or Russian ores. High content of SiO$_2$ occurs mainly in AR-1 (15.13%) and AR-3 (11.07%). On the other hand, the content of basic components is up to three times lower than for other ores. The content of adverse elements is at the standard level and does not exceed the quality requirements. The disadvantage of the iron ores used iron ores is their very low alkalinity, which directly affects the richness of the resulting agglomerate.

Another batch component is iron ore concentrate. Only one sort was used in the experiment to replace some part of the concentrate with tested iron ore (AR-5). Normally, two to three sorts of concentrate are used in a single homogenized pile at the plant. The chemical composition of the concentrate used in the experiment is shown in Table 2.

**Table 2.** Chemical composition of concentrate used in experiment.

| Concentrate | Humidity | Fe | FeO | Fe$_2$O$_3$ | SiO$_2$ | CaO | MgO | Al$_2$O$_3$ | Mn | P |
|---|---|---|---|---|---|---|---|---|---|---|
| | (%) | | | | | | | | | |
| KC1 | 9.79 | 67.95 | 27.80 | 66.16 | 4.89 | 0.12 | 0.24 | 0.16 | 0.03 | 0.011 |
| Concentrate | Basicity | S | Na$_2$O | K$_2$O | TiO$_2$ | Pb | Zn | As | C | Cl |
| | (%) | | | | | | | | | |
| KC1 | 0.070 | 0.123 | 0.029 | 0.060 | 0.015 | 0.001 | 0.001 | 0.002 | 0.152 | 0.050 |

The concentrate used has a significantly higher iron content compared to iron ores. The humidity is almost 10%, which is less efficient in terms of transport than in the case of iron ores. An advantage is that making homogenized heaps requires no additional damping. Another advantage of the concentrate used is the low SiO$_2$ content. In terms of adverse content, this concentrate is considered standard.

The last used iron-ore component was tested iron ore (AR-5), Table 3. From the chemical point of view, this iron ore has high iron content. Its main advantage is the low SiO$_2$ content, which significantly influences the alkalinity of the mixture. The basicity of this ore far exceeds the concentrates and significantly exceeds the iron ores used. Moreover, the MgO and CaO contents are the highest in comparison with the other iron ore raw materials. The negative feature is the increased sulfur content relative to standard iron ores, but even so the sulfur content is half that of other concentrates. The very low alkali content is positive.

**Table 3.** Chemical composition of the tested iron ore used in the experiment.

| Tested Iron Ore | Humidity | Fe | FeO | $Fe_2O_3$ | $SiO_2$ | CaO | MgO | $Al_2O_3$ | Mn | P |
|---|---|---|---|---|---|---|---|---|---|---|
| | (%) | | | | | | | | | |
| AR-5 | 4.78 | 62.22 | 4.45 | 84.04 | 5.54 | 0.78 | 0.57 | 1.60 | 0.06 | 0.30 |
| Tested Iron Ore | Basicity | S | $Na_2O$ | $K_2O$ | $TiO_2$ | Pb | Zn | As | C | Cl |
| | (%) | | | | | | | | | |
| AR-5 | 0.190 | 0.059 | 0.040 | 0.010 | 0.10 | 0.001 | 0.008 | 0.001 | 0.480 | - |

From the granulometric point of view, all the raw materials were sieved on 8 mm, 5.6 mm, 4 mm, 2 mm, 1 mm, 500 µm, 250 µm, 125 µm, 63 µm sieves. Although the production and technological regulations just require certain amounts of raw materials above and below 10 mm for iron ores, for the needs of these experiments it was necessary to screen the raw materials into ten fractions in order to obtain a complete view of the granulometry. A detailed analysis of the granulometric composition of the used raw materials is given in Table 4, Figure 1.

**Table 4.** Granulometric composition of used raw materials.

| Raw Materials | Above 8 mm | 5.6–8 mm | 4–5.6 mm | 2–4 mm | 1–2 mm | 500 µm$^{-1}$ mm | 250–500 µm | 125–250 µm | 63–125 µm | Under 63µm |
|---|---|---|---|---|---|---|---|---|---|---|
| | (%) | | | | | | | | | |
| Standard Iron Ores | | | | | | | | | | |
| AR-1 | 4.5 | 6.7 | 6.4 | 12.4 | 11.5 | 11.4 | 11.2 | 16.8 | 12.30 | 6.80 |
| AR-2 | 10.3 | 7.8 | 6.8 | 5.9 | 12.2 | 6.9 | 13.4 | 11.8 | 13.09 | 11.81 |
| AR-3 | 16.5 | 9.3 | 6.4 | 9.8 | 8.6 | 9.6 | 12.0 | 12.2 | 10.90 | 4.60 |
| AR-4 | 8.4 | 6.9 | 6.7 | 17.4 | 18.7 | 15.7 | 15.6 | 7.4 | 2.30 | 0.90 |
| Concentrate | | | | | | | | | | |
| KC1 | 0 | 0 | 0 | 0.1 | 0.1 | 0 | 9.5 | 8.5 | 52.4 | 29.4 |
| Tested Iron Ore | | | | | | | | | | |
| AR-5 | 3.6 | 4.6 | 5.4 | 10.9 | 9.9 | 8.1 | 9.9 | 22.2 | 13.2 | 12.2 |

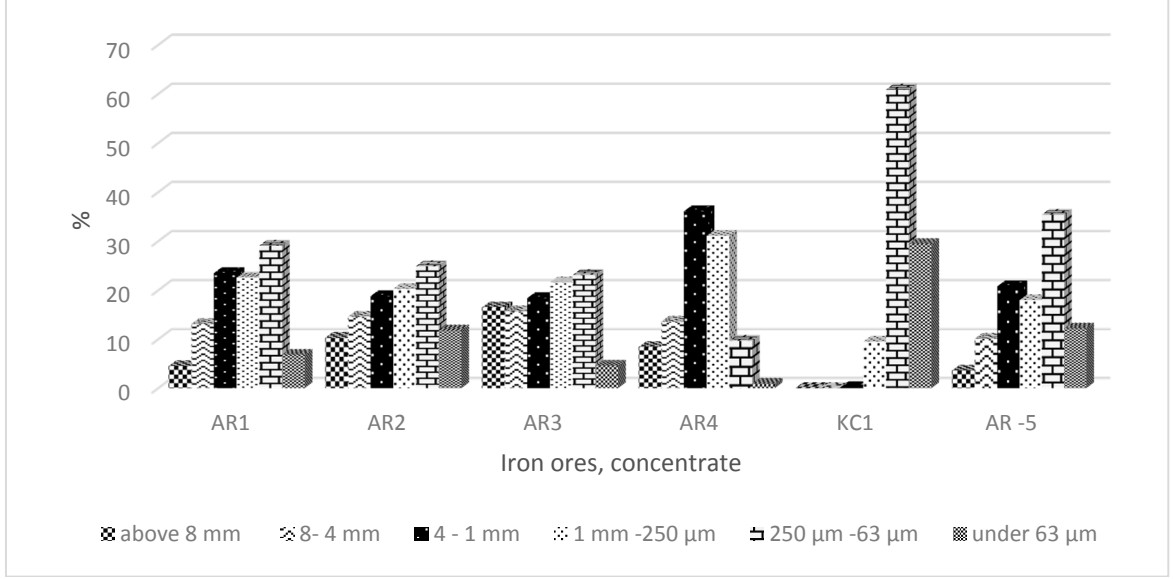

**Figure 1.** Granulometric composition of used raw materials.

From the granulometric analysis of the iron ore raw materials used in these experiments, it is evident that iron ores contain from 26% to 47% of the fraction below 500 μm. The fraction below 500 μm makes up almost 100% weight of the concentrates. In the case of tested iron ore, which formed the concentrate part in the experiments, the fraction below 500 μm made up 57.5%. Thus, it is a raw material which by the granulometric composition of its finer part is belongs among the conventional agglomerates and concentrates. Conversely, the fraction above 500 μm constitutes, in this tested iron ore, considerably smaller part than in other iron ores. Iron ores contain up to 40% of fractions above 2 mm. In the tested iron ore this fraction is less than 25%. Thus, based on screen analyses, we can say that this raw material is granulometrically on the boundary between standardly-used iron ores and standardly-used concentrates.

To study the effect of changing the ratio of concentrate to iron ore on the agglomerate phase composition, six agglomeration batches were prepared with a set percentage replacement of the concentrate by the tested iron ore AR-5. The compositions of the individual batches used in the experiments are presented in Table 5.

**Table 5.** The composition of individual experimental batches.

| Batch | A | B | C | D | E | F |
|---|---|---|---|---|---|---|
| **Concentrate Replacement by Tested Iron Ore AR-5 (%)** | **0** | **25** | **50** | **66** | **75** | **100** |
| | | | | **(%)** | | |
| *Iron Ores* | | | | | | |
| AR-1 | 5.88 | 5.84 | 5.80 | 0 | 5.76 | 5.73 |
| AR-2 | 2.35 | 2.34 | 2.32 | 0 | 2.31 | 2.29 |
| AR-3 | 16.06 | 15.96 | 15.86 | 21.31 | 15.75 | 15.65 |
| AR-4 | 11.75 | 11.68 | 11.61 | 13.70 | 11.53 | 11.45 |
| Total | 36.04 | 35.82 | 35.59 | 35.01 | 35.35 | 35.12 |
| *Concentrate* | | | | | | |
| KC1 | 36.04 | 26.86 | 17.79 | 11.91 | 8.84 | 0 |
| *Tested Iron Ore* | | | | | | |
| AR-5 | 0 | 8.95 | 17.79 | 23.1 | 26.51 | 35.12 |
| Total | 36.04 | 35.81 | 35.58 | 35.01 | 35.35 | 35.12 |
| *Other Materials* | | | | | | |
| Micropellets | 0.78 | 0.78 | 0.77 | 0.76 | 0.77 | 0.76 |
| Slag-C | 1.57 | 1.56 | 1.55 | 1.52 | 1.54 | 1.53 |
| Manganese Ore | 1.18 | 1.17 | 1.16 | 1.43 | 1.17 | 1.16 |
| Dolomite | 8.35 | 8.23 | 8.01 | 7.86 | 7.97 | 7.85 |
| Limestone | 12.64 | 13.26 | 13.94 | 15.02 | 14.48 | 15.07 |
| Coke | 3.38 | 3.38 | 3.38 | 3.38 | 3.38 | 3.38 |

The ores used in the experiments were not specially selected, they formed part of the batch for the production of agglomerate standardly used in the plant. The number of iron ores used in the experiments is comparable to commercial practice.

From the granulometric (composition) point of view, all the raw materials were sieved (screened) into several fractions. Although the production–technological regulations for iron ores state the sole condition of a certain amount of raw material above and below 10 mm, for the needs of these experiments it was necessary to screen the raw materials into ten fractions in order to obtain a complete overview of granulometry.

Based on screen analyses, therefore, we can conclude that iron ore AR-5 is granulometrically at the border between standard iron ores and commonly-used concentrates. From the point of view of the experiments needed to meet the goals of this work, it is a suitable raw material. A detailed analysis

of the granulometric composition of the raw materials used is given in Table 4, and the conditions of the experimental sintering are presented in Table 6.

The weight of one batch in the sintering pan is about 250 kg. The sucked area is 0.25 m². The height of the layer is 400 mm. The measured vacuum in individual tests during the entire experiment was from 7.0 kPa to 7.4 kPa. A constant-rpm fan was used for suction, representing an effort to maintain the same conditions in all tests.

The permeability of the mixtures was a determining factor for mixing. This was set at the 0.92 or 0.93 m·s⁻¹ for the reference mixture. In all other tests, the same permeability was achieved. Thus, the main parameter for packing raw materials into individual batches was permeability. This is also the reason for the different moisture content of individual batches. Humidity was determined by calculation. After adding the calculated amount of water, a control sample was taken from the packed mixture, and from this, the permeability of the mixture was determined. In the case of a difference in mixture permeability greater than 0.05 m·s⁻¹, additional water was added to the prepared mixture. The humidity of the mixtures ranged from 7.0% to 7.6%.

The temperature was measured along the height of the agglomeration pan at three levels, T1 max, T2 max, T3 max, while T4 max was the exhaust gas temperature, Figure 2. The thermocouples were positioned at levels 100, 200 and 300 mm from the top of the pan.

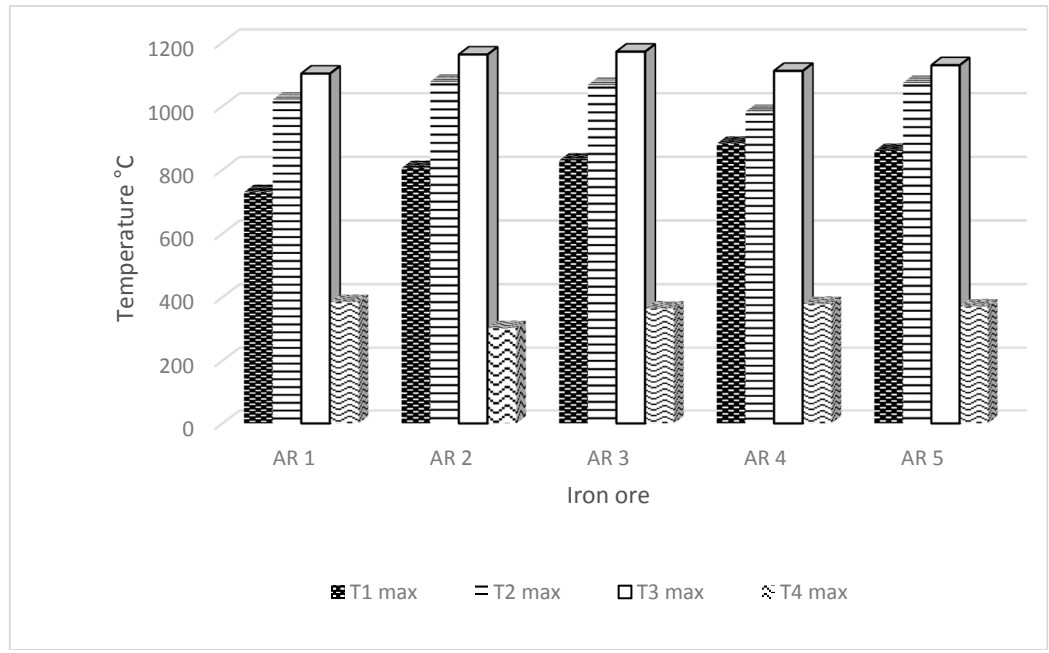

**Figure 2.** Temperature along the height of the agglomeration pan.

The experiments were carried out on a 250 kg agglomeration pan. Total dosage ranged from 235 kg to 258 kg. The differences in weights were due to the different specific weights of the concentrate portions of the mixture when the raw materials ratio was changed.

**Table 6.** Conditions of experimental sintering.

| Amount of AR-5 (%) | Moisture of Packed Mixture (%) | Batch Weight (kg) | Permeability (m·s⁻¹) | Sintering Time (min) |
|---|---|---|---|---|
| 0 | 7.40 | 253.20 | 0.92 | 62 |
| 25 | 7.40 | 253.00 | 0.91 | 52 |
| 50 | 7.20 | 248.10 | 0.92 | 44 |
| 66 | 7.00 | 237.90 | 0.92 | 37 |
| 75 | 7.60 | 249.30 | 0.90 | 42 |
| 100 | 7.60 | 235.90 | 0.93 | 31 |

Each test was repeated twice to rule out any possible measurement error during sintering. The pair-sintering measurements were averaged and the pair of sinterings was evaluated as a single sintering, whereby the variance of the values of two sinterings is minimal. Altogether, twelve sinterings, i.e., six pair tests, took place. The basicity was between 1.4 and 1.6 for all tests.

X-ray diffraction phase analysis was used to quantify the phase composition of agglomerates prepared under laboratory conditions. The samples were analysed by Seifert XRD 3003 PTS. Diffraction records were evaluated by DIFFRAC.EVA (Search-Match, KARLSRUHE & VIERSEN, Germany, company Bruker) with PDF2 and TOPAS software (version 4) using the Rietveld method. The results of the phase composition analysis are summarised in Tables 7–9.

**Table 7.** The phase composition of produced agglomerates—iron oxides.

| AR-5 (%) | Chemical Formula | $(Fe,Mg,Mn)_3O_4$ | $Fe_2O_3$ | FeO |
|---|---|---|---|---|
| | Mineralogical Name | Magnetite | Hematite | Wüstite |
| 0 | A1 | 20.20 | 43.00 | 0.85 |
| 25 | B1 | 15.80 | 42.40 | 1.05 |
| 50 | C1 | 10.70 | 37.75 | 1.15 |
| 66 | D1 | 12.60 | 33.40 | 0.75 |
| 75 | E1 | 9.60 | 42.45 | 0.90 |
| 100 | F1 | 11.20 | 37.15 | 0.90 |

**Table 8.** Phase composition of agglomerates—calcium ferrites.

| AR-5 (%) | Chemical Formula | $Ca_2Fe_2O_5$ | $Ca_4Fe_9O_{17}$ | $CaFe_5O_7$ | $Ca_2Fe_{1.89}AlO.11O_5$ | $Ca_2Fe_{22}O_{33}$ | |
|---|---|---|---|---|---|---|---|
| | Mineralogical Name | Srebrodolskit | Calcium Ferrite | Calcium Ferrite | Brownmillerit | Calcium Ferrite | |
| 0 | A1 | 1.65 | 2.85 | 3.30 | 0.80 | 4.90 | 13.50 |
| 25 | B1 | 1.30 | 3.50 | 3.20 | 1.55 | 6.10 | 15.65 |
| 50 | C1 | 2.75 | 4.55 | 2.65 | 2.00 | 18.70 | 30.65 |
| 66 | D1 | 3.05 | 2.40 | 6.50 | 1.55 | 8.40 | 21.90 |
| 75 | E1 | 2.60 | 4.00 | 3.25 | 2.15 | 15.75 | 27.25 |
| 100 | F1 | 3.45 | 3.85 | 2.65 | 3.05 | 17.60 | 30.60 |

**Table 9.** Phase composition of agglomerates—silicates.

| AR-5 (%) | Chemical Formula | $Ca_2SiO_4$ | $SiO_2$ | $CaFeSi_2O_6$ |
|---|---|---|---|---|
| | Mineralogical Name | Larnit | Quartz | Hedenbergit |
| 0 | A1 | 8.80 | 3.35 | 5.10 |
| 25 | B1 | 9.25 | 4.75 | 4.55 |
| 50 | C1 | 7.90 | 3.80 | 1.40 |
| 66 | D1 | 8.10 | 3.50 | 9.25 |
| 75 | E1 | 7.45 | 4.75 | 2.15 |
| 100 | F1 | 7.75 | 3.75 | 2.45 |

The technological conditions of agglomeration batch sintering as well as qualitative indicators of the obtained agglomerate have been published in the literature [10].

## 3. Results and Discussions

The phase composition and final structure of the agglomerate results from the course of the agglomeration process. This process takes place in the heterogeneous system of gaseous–liquid–solid phases, where the gas phase assures the course of critical processes (fuel burning, heat transfer, reduction–oxidation processes). The liquid phase, resulting from the melting of the fine-grained ore and non-ore particles and the partial melting of the coarse ore particles, enables the crystallization of magnetite, hematite, calcium ferrites and partially also of silicates. After solidification it then ensures the connection of these phases into the porous sinter-agglomerate [11,12].

Due to the chemical inhomogeneity in the microvolumes of the produced melt and the high rate of its solidification, relatively large variability occurs in the composition of the microvolumes of the agglomerates and the diversity of their microtextures. The result of this process is the iron ore agglomerate, characterized by the following phases: magnetite, wüstite, hematite, calcium ferrites, ferrocalcium olivines, pyroxenes, dicalcium silicate, tricalcium silicate, residual quartz, periclase, and free CaO [13].

The assessed agglomerate in our experiments also corresponds with this phase composition. Based on the qualitative assessment of the reference agglomerate (without addition of AR-5) and the agglomerates with increasing concentrate replacement by AR-5 iron ore, it was found that any ratio of concentrate substitution with granulometrically finer iron ore led to an increase in the quality of the final product. The agglomerate quality was assessed based on the amount of the return agglomerate, the agglomerate strength TI (+6.3 mm) and the FeO content in the agglomerate [10].

The effect of changing ratio concentrate/iron ore (C/O) ratio on the phase composition of the agglomerate is less pronounced. From the qualitative point of view, the phases in all of the studied agglomerates were identical with those in the reference agglomerate, but their quantity changed, (see Tables 7–9), which ultimately affected the mechanical and metallurgical properties of the agglomerates.

Magnetite $Fe_3O_4$ forms by crystallisation from melt or hematite reduction in solid phase. It contains isomorphic admixtures of CaO and MgO. The magnetite content had a decreasing trend as the tested iron ratio was increased. The reason for this was the change in magnetite concentration as the magnetite concentrate was replaced with hematite iron ore AR-5.

The lowest magnetite content was obtained in samples C1 and E1, from which we can assume that these agglomerates should have the best reducibility. Conversely, the worst reducibility from the examined agglomerates could be expected in the reference sample, where the magnetite content is the highest, (see Table 7).

The agglomerate matrix consists of hematite, which has a decreasing trend as the iron ore/concentrate ratio increases. The decrease of hematite content in the agglomerate structure is caused by the reaction of fine $Fe_2O_3$ grains with CaO forming a ferritic bonding phase. Hematite $Fe_2O_3$ occurs either as remnant of unreacted fragments of hematite ores, in which it forms typical granular, sparsely porous to compact aggregates or individual grains or grains clusters, whereby its crystals grow at the expense of the silicate matrix and also supress magnetite crystals, or alternatively it forms thin slats within the individual magnetite grains and arises only after the basic silicate matter has solidified by secondary oxidation of the previously—crystallized magnetite.

Wüstite is individually represented only at a low level of about 1%. It is one of the most employed indicators of agglomerate quality, while the higher proportion of FeO is associated with higher strength and worse reducibility. It follows from the above that most of the FeO in the studied agglomerates was bound up in magnetite. If we consider, based on these facts, that the FeO content in the agglomerate is directly related to the magnetite content, we can assign the same significance to $Fe_3O_4$ as to FeO.

Besides magnetite and hematite, minerals based on calcium ferrites and silicates were identified in the structure of the agglomerate (see Tables 8 and 9). Calcium ferrites already begin to form at temperatures 500 to 600 °C. They occur in places with a hematite base, i.e., in cracks and pores, through which the gas phase flows. The sintering reactions usually begin in the solid state between small particles of limestone and ore grains in close contact, which then fuse together. Subsequently there occurs congruent melting of the products accompanied with the assimilation of the nucleus parts. The period during which the temperature does not drop below 1100 °C is the time required for melt formation during sintering [14].

With increasing temperature, the primary melt dissolves the gangue component and hematite to form complex compounds, silicoferrites of calcium and aluminum (SFCA). The basic component is hemicalciumferrite $CaO_2$ $Fe_2O_3$, in which $SiO_2$ and $Al_2O_3$ are isomorphically admixed. In the literature this phase is referred to as the SFCA ferrite [15].

As the percentage of concentrate replacement with iron ore AR-5 increased in our experiments, the content of the ferritic phase changes. The lowest value was reached in the standard agglomerate. In agglomerates with 50%, and 100% substitution, it achieved almost identical values of 30.65%, and 30.60%. The agglomerate with 25% substitute was the closest to the standard agglomerate, and the agglomerates with 66% and 75% substitution were in the range from 21% to 27%. However, it can be stated that the replacement of the concentrate by AR-5 in all percentages increases the number of ferritic phases in the agglomerate (see Table 8)

Formation of calcium ferrites is impeded by the presence of $SiO_2$, with which CaO reacts more readily, forming slag silicate melts. $SiO_2$ already begins to react with magnetite at 650 °C in solid state, producing fayalite. The intense reaction occurs above 1100 °C. Yet a small amount of fayalite produced by the solid-state reaction causes melting of the batch, because above the liquidus line there is an unlimited solubility between fayalite and magnetite. Fayalite may also form in hematite batches, but only at higher temperatures when hematite decays into magnetite. The products of iron oxide reaction with $SiO_2$ in the Ca-Si-Fe-O system are therefore olivines (see Table 8). The amount of calcium ferrites and dicalcium silicate phases depends on the basicity of the agglomeration mixture; with lower basicity glassy silicates and secondary hematite prevail, as well as magnetite, and calcium ferrites and dicalcium silicates phases at the higher end [16]. The basicity at which our sinterings were performed was set in the range from 1.4 to 1.6. This range was chosen because of the formation of dicalcium silicate in the basicity range from 1.6 to 1.8, causing the agglomerate to become brittle, which could result in distortion of the results.

The silicate components in the agglomerates are ferro-calcium olivines, pyroxins, calcium silicates, and solid glassy silicates. Ferro-calcium olivines $(Ca_XFe_{1-X})_2 \cdot SiO_4$ form solid solutions between fayalite $2FeO \cdot SiO_2$ and dicalcium silicate $2CaO.SiO_2$. The phase composition of silicate phases in studied agglomerates is presented in Table 9.

As can be seen from Table 9, the most represented phase of silicate phases is $Ca_2SiO_4$ (larnite).

The amount of larnite in the agglomerate depends on the agglomerate basicity. With increasing basicity, its amount increases. In the area of basicity of about 1.8, which corresponds to the $Ca_2SiO_4$ molecule, the agglomerate loses its strength (decays). The disintegration of the agglomerate is attributed to the polymorphic conversion of $Ca_2SiO_4$, which is associated with an increase in volume of about 12%. For this reason, a lower level of basicity was chosen in the experiment, i.e., 1.4 to 1.6, where although larnite is present, its amount ranging from 7.45% to 9.25% is so small that it does not cause the agglomerate to break down and thus does not affect its strength [17,18].

The results of the experimental study show that in all agglomerates, in qualitative sense, present phases were identical to the reference agglomerate, only their intensity was changed. The change in phase intensity did not reduce the qualitative parameters of the compared agglomerates.

## 4. Conclusions

Based on the theoretical knowledge from the study of the structure and phase composition of the agglomerate, the latter can be defined as a multiphase complex whose properties depend on the features and volume concentration of the individual phases and their relative distribution. Furthermore, based on the knowledge of the properties of the individual phases, the nature of their impact on the properties of the final agglomerate can be generally assessed.

On the basis of the results obtained from this study of the phase composition of a conventional agglomerate and agglomerate with changed concentrate/iron ore ratio it can be stated that no new phases occur, and only the intensity of the individual phases changes.

In the standard agglomerate and agglomerate with 25%, 50%, 66%, 75% and 100% replacement of concentrate by the iron ore AR-5, the following phases were identified by X-ray analysis: magnetite, wüstite, hematite, calcium ferrites, ferro-calcium olivines, pyroxenes, dicalcium silicate, residual quartz.

Replacement of magnetite concentrate with hematite ore significantly reduces the content of magnetite. While in standard agglomerate, its quantity is 20.2%, its share in the agglomerate with the

AR-5 iron ore substitution ranges from 9.6% to 15.8%. In the agglomerate without concentrate, the amount of magnetite dropped to 11.2%, which is almost a 50% decrease.

The change in the amount of hematite is not so pronounced. In a standard agglomerate the amount of hematite corresponds to 43%, while in other compared agglomerates, it is ranging from about 33% to 42%. A slight decrease in hematite is probably due to the reaction of fine grains of $Fe_2O_3$ with CaO forming a ferritic binding phase.

The amount of ferritic phases changes significantly. While the standard agglomerate reaches 13.5%, the agglomerates with hematite ore AR-5 reach values in the range of about 15% to 30% of the ferritic phases.

Wüstite content is at the level of about 1%, and it is assumed that most of it is bound in magnetite in the agglomerates.

The silicate phases are represented by ferro-calcium olivines, larnite and residual quartz. The amount of silicate phases stays within a narrow range. Of these silicate phases, larnite deserves the most attention, as it undergoes polymorphic transformation during cooling which is accompanied by an increase in volume by about 12%, and as a result, the agglomerate breaks down, leading to the loss of strength. The amount of larnite in all compared agglomerates ranged between 7% and 9%. The amount of larnite is small and it is therefore assumed that it becomes is physically stabilized in the ferritic binding phase and does not affect the agglomerate's strength.

Based on the results of the phase composition of the compared agglomerates, it can be stated that the matrix of agglomerates with the changed ratio of concentrate/ore AR-5 is formed by hematite, and the binding phase is mainly composed of ferritic phases and, to a lesser extent, silicate phases. Such a phase composition, in terms of quality, ensures good metallurgical properties. These agglomerates are easily reducible, which also corresponds to lower consumption of metallurgical coke in the process of pig iron production.

High quality agglomerate for blast-furnace processing must therefore consist of mineral phases guaranteeing a good strength of and its easy reducibility, both of which were achieved in this study. Based on the qualitative assessment of the reference agglomerate (without the addition of AR-5) and agglomerates with increasing concentrate replacement by iron ore AR-5, it can be stated that any ratio of replacement of the concentrate by granulometrically finer iron ore leads to an increase in the quality of the final product.

This research has shown that also in the case of replacement of the concentrate with ore approaching the properties of the concentrate in its composition and granulometry it is possible to ensure, in the sintered layer, conditions for the formation of mineral phases in such ratio that there is no great degradation of the properties of the finished agglomerate. In terms of the main objective, which is the quality of agglomerate and reduction of the environmental burden from agglomeration plants, change in the C/O ratio or concentrate replacement with the above-defined ore may be recommended for the iron ore agglomerate production process as the basic blast furnace batch component.

**Author Contributions:** M.F. performed the methodology, investigation, conceptualization, writing original draft preparation, writing review and editing; D.I. investigation, formal analysis, resources; R.F. investigation, validation; M.D. investigation, visualization, project administration; J.L. investigation, writing review and editing.

**Funding:** This research was funded by [APVV] Slovak Research and Development Agency, Slovak Republic number APVV–16-0513 and [VEGA MŠ SR a SAV] grant number 1/0847/16.

**Conflicts of Interest:** The authors declare no conflicts of interest.

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
