# Peer review of "The Effect of Concentrate/Iron Ore Ratio Change on Agglomerate Phase Composition"

_metals, doi:10.3390/met8110973_

Round 1
Reviewer 1 Report
This paper is interesting and bring a real amount of new result about changing the ratio of concentration and iron ore. On the other hands, there are several weakness points for publish.
First, experimental condition is not clear. Especially, the relationship between this paper and reference 8 must be make clear. I thought they are same experiment and analyze of sinter was added in this paper. However, FeO content shown in Table 6 is different from that shown in Table 2 of reference 8. This means thy are different experiment or FeO is not wustite just Fe2+(but in this case, total Fe2+ would be also different) please explain the relation clearly.
Second, in conclusion, quality of sinter was just discussed by phase of sinter. In reference 8, shatter test was carried out and the result is significant for quality of sinter. Then, relationship between the result of this paper about phase and the result of reference 8 should be important for understanding the quality of sinter. Utilizing the result, general information would not need at least strength part.
Please find below some further remarks.
In total all parts of this paper, concentrate/iron was shown as “C/O” in Abstract and “CO/O” in text. That makes reader confuse. Term should be uniformed.
Table 1 - 3 would be summarized in one table. if there is reason not to summarize, it should be keep three tables.
P1 L30 In introduction, at first paragraph, the text is generally correct but reference should be written.
P2 L65 In this paragraph, reference should be written.
P2 L85 in this experiment, origin of tested-ore, AR-5, should be written. (AR-5 is just produced raw material or treated some operation like grinding) it should be written for reproducibility within explainable range.
P2 L93 in Table 1, “Iron ores” and “Iron Ores” are used. It should be uniformed.
P3 L111 in P2 L 87 AR-5 was explained as “high-iron ore” and in this part it was “tested iron ore”. It should be uniformed.
P4 L119 in this paragraph, granulometric composition was shown. For easy understanding, graph of each ores would help reader.
P5 L141 Basic experimental condition such as diameter and height of sintering pot, height of sample, suction method, moisture, diameter of sample and so on should be explained in this paper even if it was written in reference [8].
Especially characteristic of mixture is important for sinter. In this case, mixing ratio of concentrate to tested-iron ore was changed. Then, optimized moisture has possibility to be changed for granulation and strength. The reason for same moisture should be written.
P5 L141 in this experiment, for the main target shown in introduction, it is difficult to understand that the reason why mixing ratio of AR-1 to 4 was changed. If composition of mixture would be kept, basicity should be uniformed. Then, please explain the reason why mixing ratio of mixture A to F and basicity were changed.
P5 L148 by reading reference [8], maximum temperature and heat pattern was not clear. They are significantly important to understanding the sintering condition. It should be clear.
P5 L181 “The magnetite content has a decreasing trend.” would need “with increasing the tested iron ratio.”. And the reason of this change was not clear by explanation “The reason for this is …”. because the magnetite content was not completely depended on the ratio of AR-5. It should be explained in reasonable.
P5 L184 “we can assume …” of course, generally, low magnetite content leads high reducibility if other conditions are same. In this case, other factors, such as strength, porosity, composition and so on, are not clear. This assumption would be led with other evidence or explain with at least reference.
P6 L200 If FeO content in samples are changed with tested-iron content, the assign would be reasonable, however, it was not changed with the ratio of AR-5. It is difficult to understand. The explanation of FeO content should be made more detail.
P6 L203 P6 L211 these paragraphs are explaining general information and not have relation with your results. They are not result and discussion. Please make clear relation with your result.
P6 L204 There are no Table 20 and 21. This would be miss. Or clear explanation is necessary.
P6 L216 “reaches about 30% ” would change to 30% of what.
P6 L216 As shown in Table 7, the content of ferritic phases was not reaches 30% in D1 and E1. The explanation “agglomerate with over 25 % AG-5 reaches about 30 %. ”is not collect. This result should be explain more collect and detail.
And “over 25%” conclude 30%, 40% and so on. From this result, it is not clear that ferritic phases ratio shows high value in these cases. Sentence would be corrected.
P7 L219 Is this paragraph confirmed by yourself or generally information? It is not clear. This paragraph is difficult to understand the purpose of text.
P7 L221 In order to discuss the phase in sinter, temperature history is important information. Without the information, reader can guess temperature as general sintering process but it is just conjecture. In this test, It is not clear in this text. Please explain it. In addition, position of these phase in sinter product is important. If it was analyzed, it would assist the discussion in this paragraph.
P7 L231 Sorry, “This range was …”this sentence is too complex to understand. What was the “distortion of the result”? Please explain it. And if the “distortion of the result” means the Fe3O4 ratio and FeO ratio, the logic of these result, which caused by the basicity, must be explained in detail. In P7 L240 “Larnite is one of the most important minerals in terms of agglomerate strength.” In P7 L239 “Since the strength of the agglomerate had a significantly different course from the larnite content” They are seem to be inconsistent. I can’t understand the purpose of these sentence. Please explain in detail with reference.
P7 L259 “the change in phase …” Where did you explain this result in this paper? The result shows produced phases. Physical property was tested in reference 8 and it should be written as it is. And reduction test was not carried out in this text then it should be written as assumption. If it was carried out in other paper, write them in correct.
P7 L260 “In the case he …” I guess this is miss spell of “the”. “requirements of industrial particles” was not shown in this text and reader can’t judge it meet or not please explain definition with reference.
Author Response
(Reviewer 1)
Yes | Can be improved | Must be improved | Not applicable | |
Does the introduction provide sufficient background and include all relevant references? | (x) | ( ) | ( ) | ( ) |
Is the research design appropriate? | ( ) | (x) | ( ) | ( ) |
Are the methods adequately described? | (x) | ( ) | ( ) | ( ) |
Are the results clearly presented? | ( ) | ( ) | (x) | ( ) |
Are the conclusions supported by the results? | ( ) | ( ) | (x) | ( ) |
Comments and Suggestions for Authors
This paper is interesting and bring a real amount of new result about changing the ratio of concentration and iron ore. On the other hands, there are several weakness points for publish.
First, experimental condition is not clear. Especially, the relationship between this paper and reference 8 must be make clear. I thought they are same experiment and analyze of sinter was added in this paper. However, FeO content shown in Table 6 is different from that shown in Table 2 of reference 8. This means thy are different experiment or FeO is not wustite just Fe2+(but in this case, total Fe2+ would be also different) please explain the relation clearly.
Experimental conditions are the same in both articles. The article (now ref. 10) was focused on the influence of concentrate / iron ore ratio on agglomerate quality and sintering conditions. This article focuses on the effect of the concentrate / iron ore ratio on the agglomerate phase composition.
The FeO content (now ref. 10) Table2 was determined by chemical analysis
The FeO content (now Table 7)was determined by X-ray phase analysis; it is a separate phase of FeO (wustite)
Second, in conclusion, quality of sinter was just discussed by phase of sinter. In reference 8, shatter test was carried out and the result is significant for quality of sinter. Then, relationship between the result of this paper about phase and the result of reference 8 should be important for understanding the quality of sinter. Utilizing the result, general information would not need at least strength part.
The quality of the agglomerate (now ref. 10) was assessed mainly on the basis of the strength, which is the basic condition for the blast furnace batch.
In this paper, attention was focused on the quality of the agglomerate in terms of phase composition. The replacement of concentrate by agglomeration ore AR5 reduced the content of magnetite and increased the content of hematite, which had a significant impact on agglomerate reducibility. Magnetite is difficult to reduce, which is related to higher fuel consumption.
Please find below some further remarks.
In total all parts of this paper, concentrate/iron was shown as “C/O” in Abstract and “CO/O” in text. That makes reader confuse. Term should be uniformed.
Correct is C/O
Table 1 - 3 would be summarized in one table. if there is reason not to summarize, it should be keep three tables.
P1 L30 In introduction, at first paragraph, the text is generally correct but reference should be written.
P2 L65 In this paragraph, reference should be written.
P2 L85 in this experiment, origin of tested-ore, AR-5, should be written. (AR-5 is just produced raw material or treated some operation like grinding) it should be written for reproducibility within explainable range.
AR-5 is just produced raw material (as delivered)
P2 L93 in Table 1, “Iron ores” and “Iron Ores” are used. It should be uniformed.
Iron ores
P3 L111 in P2 L 87 AR-5 was explained as “high-iron ore” and in this part it was “tested iron ore”. It should be uniformed.
tested iron ore
P4 L119 in this paragraph, granulometric composition was shown. For easy understanding, graph of each ores would help reader.
P5 L141 Basic experimental condition such as diameter and height of sintering pot, height of sample, suction method, moisture, diameter of sample and so on should be explained in this paper even if it was written in reference [8].
Especially characteristic of mixture is important for sinter. In this case, mixing ratio of concentrate to tested-iron ore was changed. Then, optimized moisture has possibility to be changed for granulation and strength. The reason for same moisture should be written.
The ores used for the experiment were not specially selected, they made part of the batch for the production of the agglomerate used in the plant. The number of iron ores used for the experiment is comparable to practice.
From the granulometric (composition) point of view, all the raw materials were sieved (screened) into several fractions. Although the production-technological regulations for iron ores state only a condition of a certain amount of raw material above and below 10 mm, for the needs of this experiment it was necessary to screen the raw materials into ten fractions in order to obtain a complete overview of granulometry.
Based on screen analyzes, we can therefore conclude that iron ore AR-5 is granulometrically at the border between standard iron ores and standardly used concentrates. From the point of view of the experiments needed to meet the goals of this work, it is a suitable raw material. A detailed analysis of the granulometric composition of the raw materials used is given in Table 4 and the conditions of the experimental sintering are in Table 6.
The weight of one batch in the sintering pan is about 250 kg. The sucked area is 0.25 m2. The height of the layer is 400 mm. The measured vacuum at individual tests of the entire experiment was from 7.0 kPa to 7.4 kPa. A constant-rpm fan was used for suction, so there was an effort to maintain the same conditions for all tests.
The permeability of the mixtures was a determining factor for mixing. This was set at the 0.92 or 0.93 m.s-1 for the reference mixture. In all other tests, the same permeability was achieved. Thus, the main parameter for packing raw materials into individual batches was permeability. This is also the reason for the different moisture content of individual batches. Humidity was determined by calculation. After adding the calculated amount of water, a control sample was taken from the packed mixture. From this, the permeability of the mixture was determined. In the case of a difference in permeability of the mixture that was greater than 0.05 m.s-1, additional water was added to the prepared mixture. The humidity of the mixtures ranged from 7.0 % to 7.6 %.
Table 6. The conditions of experimental sintering
Amount of AR-5 | Moisture of packed mixture | Batch weight | Permeability | Sintering time |
(%) | (%) | (kg) | (m.s-1) | (min) |
0 | 7,40 | 253,20 | 0,92 | 62 |
25 | 7,40 | 253,00 | 0,91 | 52 |
50 | 7,20 | 248,10 | 0,92 | 44 |
66 | 7,00 | 237,90 | 0,92 | 37 |
75 | 7,60 | 249,30 | 0,90 | 42 |
100 | 7,60 | 235,90 | 0,93 | 31 |
P5 L141 in this experiment, for the main target shown in introduction, it is difficult to understand that the reason why mixing ratio of AR-1 to 4 was changed. If composition of mixture would be kept, basicity should be uniformed. Then, please explain the reason why mixing ratio of mixture A to F and basicity were changed.
Because of the different SiO2 content in the agglomeration mixture, the total content of acidic and basic constituents varied, with basicity ranging from 1.4 to 1.6.
In the experiments, were used 250 kg batches of agglomeration mixture, and to maintain the basicity at a preset value was very difficult.
P5 L148 by reading reference [10], maximum temperature and heat pattern was not clear. They are significantly important to understanding the sintering condition. It should be clear.
The temperature was therefore not recorded, but it was measured along at the height of the agglomeration pan at three levels: T1 max, T2 max, T3 max, T4 exhaust 400 °C
P5 L181 “The magnetite content has a decreasing trend.” would need “with increasing the tested iron ratio.”. And the reason of this change was not clear by explanation “The reason for this is …”. because the magnetite content was not completely depended on the ratio of AR-5. It should be explained in reasonable.
The magnetite content has a decreasing trend with increasing the tested iron ratio. The reason for this is a concentration change of magnetite, because the magnetite concentrate is replaced by hematite iron ore AR-5.
P5 L184 “we can assume …” of course, generally, low magnetite content leads high reducibility if other conditions are same. In this case, other factors, such as strength, porosity, composition and so on, are not clear. This assumption would be led with other evidence or explain with at least reference.
In the experiment, the same conditions were observed for all sinterings. Only the C/O ratio was varied and only the change in phase composition was monitored. I agree with the opinion that there are other factors that affect the physical and physical-chemical properties of the agglomerate, i.e. its quality.
P6 L200 If FeO content in samples are changed with tested-iron content, the assign would be reasonable, however, it was not changed with the ratio of AR-5. It is difficult to understand. The explanation of FeO content should be made more detail.
By changing the O/C ratio, the amount of FeO in the mixture decreases. Wustite as a separate phase is represented only at a low level of about 1 %, the probable cause is that it is part of other compounds, namely magnetite or calcium ferrites or silicates.
P6 L203 P6 L211 these paragraphs are explaining general information and not have relation with your results. They are not result and discussion. Please make clear relation with your result.
I agree that this is general information
P6 L204 There are no Table 20 and 21. This would be miss. Or clear explanation is necessary.
P6 L216 “reaches about 30 % ” would change to 30 % of what.
ferritic phases
P6 L216 As shown in Table 7, the content of ferritic phases was not reaches 30% in D1 and E1. The explanation “agglomerate with over 25 % AG-5 reaches about 30 %. ”is not collect. This result should be explain more collect and detail.
And “over 25%” conclude 30%, 40% and so on. From this result, it is not clear that ferritic phases ratio shows high value in these cases. Sentence would be corrected.
With the increasing percentage of concentrate replacement by iron ore AR-5, the content of the ferritic phases changes. The lowest value is reached in a standard agglomerate. In agglomerates with 50 %, and 100 % substitution, it achieves almost identical values of 30.65 %, and 30.60 %. The agglomerate with a 25 % substitute is the closest to the standard agglomerate, and the agglomerates with 66, and 75 % substitution are in the range of 21 to 27 %. However, it can be stated that the substitution of the concentrate by AR-5 in all percentages increases the amount of ferritic phases in the agglomerate.
P7 L219 Is this paragraph confirmed by yourself or generally information? It is not clear. This paragraph is difficult to understand the purpose of text.
It is generally information
P7 L221 In order to discuss the phase in sinter, temperature history is important information. Without the information, reader can guess temperature as general sintering process but it is just conjecture. In this test, It is not clear in this text. Please explain it. In addition, position of these phase in sinter product is important. If it was analyzed, it would assist the discussion in this paragraph.
I agree that temperature plays an important role in shaping the structure of the agglomerate, and its course should be recorded. In the conditions, under which the experiment was performed, it was not possible. The temperature was therefore not recorded, but it was measured along at the height of the agglomeration pan at three levels.
P7 L231 Sorry, “This range was …”this sentence is too complex to understand. What was the “distortion of the result”? Please explain it. And if the “distortion of the result” means the Fe3O4 ratio and FeO ratio, the logic of these result, which caused by the basicity, must be explained in detail.
Because of the different SiO2 content in the agglomeration mixture, the total content of acidic and basic constituents varied, with basicity ranging from 1.4 to 1.6.
In the experiments, were used 250 kg batches of agglomeration mixture, and to maintain the basicity at a preset value was very difficult.
In P7 L240 “Larnite is one of the most important minerals in terms of agglomerate strength.” In P7 L239 “Since the strength of the agglomerate had a significantly opposite course from the larnite content” They are seem to be inconsistent. I can’t understand the purpose of these sentence. Please explain in detail with reference.
The amount of larnite in the agglomerate is dependent on the agglomerate basicity. With increasing basicity, its amount increases. In the area of basicity of about 1.8, which corresponds to the Ca2SiO4 molecule, the agglomerate loses its strength (decays). The disintegration of the agglomerate is attributed to the polymorphic conversion of Ca2SiO4, which is associated with an increase in volume of about 12 %. For this reason, a lower level of basicity was chosen in the experiment, i.e. 1.4 to 1.6, where although the larnite is present, its amount ranging from 7.45 to 9.25 is so small that it does not cause the agglomerate to break down and thus does not affects its strength.
P7 L259 “the change in phase …” Where did you explain this result in this paper? The result shows produced phases. Physical property was tested in reference 8 and it should be written as it is. And reduction test was not carried out in this text then it should be written as assumption. If it was carried out in other paper, write them in correct.
The result of this paper is, that we can state, that that concentrate substitution by iron ore with suitable granulometry is possible and that this substitution in any O/C ratio does not cause a decrease in the physical and physicochemical properties of the agglomerate. This statement is based on analysis of the phase composition of the agglomerates.
P7 L260 “In the case he …” I guess this is miss spell of “the”. “requirements of industrial particles” was not shown in this texcont and reader can’t judge it meet or not please explain definition with reference.
Thank you very much for the review of our article. We appreciate and accept your comments. The comments are incorporated into the text in the attachment as "Reviewer 1".
Thank you very much for your cooperation and comments.
Best regards,
Author's team (Mária Frohlichová)

Reviewer 2 Report
see attachment.

Author Response
Thank you very much for the review of our article. We appreciate and accept your comments. The comments are incorporated into the text in the attachment as "Reviewer 2".
Thank you very much for your cooperation and comments.
Best regards,
Author's team (Mária Frohlichová)

Reviewer 3 Report
In this work, the influence of the ratio of concentrate to iron ore on the phase composition of the iron ore agglomerate has been studied. Before the further consideration the follwoing issues should be addressed:
The novelty of the work is missing and should be highlighted.
The reported chemical composition of iron ores and concentrates has been analysed in this work or is reported according to the datasheets!
The first paragraph of results and discussion should be moved to the materials and methods section.
It is recommended to insert the X-ray diffraction patterns!
Author Response
The novelty of this paper is, that we can state, that concentrate substitution by iron ore with suitable granulometry is possible and that this substitution in any O/C ratio does not cause a decrease in the physical and physicochemical properties of the agglomerate. This statement is based on analysis of the phase composition of the agglomerates.
The reported chemical composition of iron ores and concentrates has been analysed in this work.
The first paragraph of results and discussion should be moved to the materials and methods section. I agree.
Thank you very much for the review of our article. We appreciate and accept your comments. The comments are incorporated into the text in the attachment as "Reviewer 3".
Thank you very much for your cooperation and comments.
Best regards,
Author's team (Mária Frohlichová)

Round 2
Reviewer 1 Report
Thank you revise first version in detail.
Some small revise should be carried out for publish.
In Table 5, C/O ratios are same in all samples. This would be mistake. Please revise.
Position T1 to T3 should be written clearly.
Author Response
The temperature was measured along the height of the agglomeration pan at three levels, T1 max, T2 max, T3 max, while T4 max was the exhaust gas temperature, Fig. 2. The thermocouples were positioned at levels 100, 200 and 300 mm from the top of the pan.
Thank you very much for review.

Reviewer 3 Report
This paper can be accepted in this form.
Author Response
Thank you very much for rieview. The artical was english corrected.
